# Learning to ignore: Single Source Domain Generalization via Oracle Regularization

## Abstract

Machine learning frequently suffers from the discrepancy in data distribution, commonly known as domain shift. Single-source Domain Generalization (sDG) is a task designed to simulate domain shift artificially, in order to train a model that can generalize well to multiple unseen target domains from a single source domain. A popular approach is to learn robustness via the alignment of augmented samples. However, prior works frequently overlooked what is learned from such alignment. In this paper, we study the effectiveness of augmentation-based sDG methods by analyzing the data generating process. We highlight issues in using augmentation for OOD generalization, namely, the distinction between domain invariance and augmentation invariance. To alleviate these issues, we introduce a novel regularization method that leverages pretrained models to guide the learning process via a feature-level regularization of mutual information, which we name PROF (Progressive mutual information Regularization for Online distillation of Frozen oracles). PROF can be applied to conventional augmentation-based methods to moderate the stochasticity of models repeatedly trained on augmented data. We show that PROF stabilizes the learning process for sDG.

## 1 Introduction

Distribution shift is prevalent in many machine learning settings. The term is often referred to as *domain shift*, where a domain is understood as the joint probability distribution from which samples are drawn. An important aspect of domain shift is that it severely hinders the generalizability of trained models (Kurakin et al., 2018). The issue is easily observable when a model trained in a source domain suffers in a target domain that is inconsistent with the source. Domain Generalization (DG) is a task specifically devised to test a model's robustness under domain shift, where the model is given multiple labeled datasets at training time (Gulrajani & Lopez-Paz, 2021). Single-source Domain Generalization (sDG) is a variant of DG, where only a single source domain is provided at train time. The absence of additional source domains makes sDG challenging, mainly because conventional DG methods that leverage multiple domains cannot be easily adopted. To overcome such barriers, prior works on sDG often utilize data augmentation to generate unseen domains (Volpi et al., 2018) and learn domain-invariant features through an alignment of the generated domains using self-supervised contrastive loss (Oord et al., 2018) (hereinafter contrastive loss).

However, there is a relative void in the discussion on what is learned through the alignment of augmented samples. In this paper, we analyze the effectiveness of augmentation-based sDG approaches from a novel perspective of style-content disentanglement. Style-Content (S-C) disentanglement aims to identify a partitioned latent space, namely style, and content (Ren et al., 2021; Hyvarinen & Morioka, 2016). While the definitions of style and content vary across settings, here we define content as latent features that are invariant across augmentations (i.e. augment-invariant), while style is the latent feature subpart that changes with the augmentation. Recently, Von Kügelgen et al. (2021) studied an interesting connection between S-C disentanglement and data augmentation, demonstrating that contrastive learning provably learns to retrieve the augment-invariant features under some assumptions. We connect the discovery to the sDG literature to analyze the effectiveness of retrieving domain-invariant information from augmented data. We examine the problem from a causal standpoint by illustrating it via a causal graph (Pearl, 2009).

We state our contributions as follows. (1) We analyze the single source domain generalization task through the lens of S-C disentanglement and highlight the difficulties of learning domain-invariant information from augmentation-based sDG methods. (2) We empirically show that augmentation-based sDG methods display large fluctuations in OOD performance across various datasets (3) To mitigate the issues brought by the aforementioned obstacles, we introduce a novel regularization method PROF for sDG. (4) We further devise a novel alignment objective MDAR (Multi-Domain Alignment with Redundancy reduction) that serves as a strong sDG baseline.

## 2 PRELIMINARIES

Learning domain agnostic models from limited source domains is a longstanding area of investigation. In this section, we revisit related works on S-C disentanglement and domain generalization.

**Style-Content Disentanglement**  S-C disentanglement seeks to separate the aggregated latent variable into two parts, denoted as style and content. While the term style and content originated from the style transfer literature (Mathieu et al., 2016; Szabó et al., 2017), recent works try to push the idea further using concepts of causal inference (Pearl, 2009; Peters et al., 2017) and Independent Component Analysis (ICA) (Locatello et al., 2018; Gresele et al., 2021; Reizinger et al., 2022). Notably, disentanglement is used to elucidate the underlying mechanism of data augmentation (Von Kügelgen et al., 2021; Ilse et al., 2021; Huang et al., 2022; Mitrovic et al., 2021).

**Domain Generalization**  In the multi-source domain generalization field, disentanglement of domain-invariant features has shown great success in training robust domain-agnostic models by leveraging shared information across domains. To learn domain-invariant information, researchers commonly analyze the data generating process (DGP) using structural causal models to design effective algorithms (Arjovsky et al., 2019; Mahajan et al., 2021; Wang & Veitch, 2022). On the contrary, disentanglement is rarely discussed in the sDG literature. This is due to innate conditions of sDG, where only one domain is available for training. This setting makes it hard to apply conventional disentanglement approaches developed in the multi-DG literature. To tackle this, a line of work focuses on how to augment *unseen* domains effectively with generative models (Volpi et al., 2018; Qiao et al., 2020; Li et al., 2021; Wang et al., 2021; Wan et al., 2022; Fan et al., 2021). However, there is a lack of discussion on whether augmented samples can simulate unseen domains, or whether it can be used to learn domain-invariance. A recent movement in the multi-DG literature highlights the use of pretrained models for OOD generalization, leveraging the knowledge of the pretrained models (Cha et al., 2022; Wortsman et al., 2022; Li et al., 2023). Such works closely resemble the methods introduced in the Knowledge Distillation (KD) literature (Hinton et al., 2015; Adriana et al., 2015; Ahn et al., 2019; Shrivastava et al., 2023; Tian et al., 2020).

## 3 LIMITATIONS OF AUGMENTATION FOR sDG

In this section, we present an overlooked problem of augmentation-based sDG methods. Specifically, we revisit recent works on S-C disentanglement to analyze the effectiveness of utilizing augmentation for out-of-domain generalization.

**A general view towards augmentation-based sDG methods**  We present a general expression for augmentation-based sDG methods and discuss their effectiveness. Generally, augmentation-based methods can be expressed as *augment and align*, minimizing the following objective (omitting some arguments for simplicity) denoting $x$ and $\bar{x}$ as an original sample and its augmented view:

$$L := L_{ce} + L_{\text{MaxEnt}}(x, \bar{x}; \Phi). \tag{1}$$

where $L_{ce}$ is the cross-entropy loss $L_{ce}(\mathbf{y}, \hat{\mathbf{y}}) = -\sum_i y_i \log(\hat{y}_i)$ with $\mathbf{y}$ the ground truth, $\hat{\mathbf{y}}$ the softmax prediction of the model, and $L_{\text{MaxEnt}}$ is an objective that simultaneously aligns the mapped representations $\Phi(x)$ and $\Phi(\bar{x})$ under entropy regularization, where $\Phi$ is a feature extractor. Commonly, contrastive loss is used as $L_{\text{MaxEnt}}$. Recently, Von Kügelgen et al. (2021) showed that the optimization of a contrastive loss provably minimizes $L_{\text{MaxEnt}}$, learning $\Phi$ to extract features that are augment-invariant, under a certain condition. In this perspective, conventional augmentation-based sDG methods could be understood as retrieving augment-invariant features.

**A causal interpretation of data augmentation**    We illustrate the underlying data generating process (i.e., DGP) using a causal graph and incorporate data augmentation into the causal graph under the sDG setting. An instance of a given labeled dataset is typically composed of an observation $X$ (i.e., image) and its label $Y$. Although supervised learning predicts $Y$ directly from $X$, this does not reflect the underlying causality. We can think of the existence of hidden features (e.g., real-world attributes regarding the subject of the image and the background), which we will refer $W$, that affect both the image and label. At this moment, the causal graph for DGP can be simply represented as $X \leftarrow W \rightarrow Y$ where $W$ is unobserved.

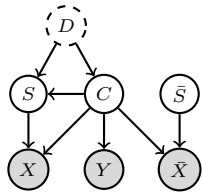

Figure 1: A causal diagram depicting DGP under data augmentation.

Now, we incorporate data augmentation into the picture. Given *label-preserving* augmentation methods, we attain $\bar{X}$ the augmented view of $X$. Such an augmentation can be considered as manipulating only the style $S$ (augment-variant) to yield $\bar{S}$ while retaining its content (augment-invariant) $C$ where $C$ and $S$ partitions $W$, that is, $W = (C, S)$ (see Von Kügelgen et al. (2021) for a detailed discussion). Yet, this does not imply that $C$ and $S$ are independent. $C$ causally affects $S$ (also corroborated by experimental results (Klindt et al., 2021)). A way to understand this separation is by viewing such an augmentation as a soft intervention (Eberhardt & Scheines, 2007) on $S$, resulting in a modified style $\bar{S}$. By definition, $(C, \bar{S})$ becomes the hidden features of $\bar{X}$. Furthermore, $C$ consistently affects $Y$ regardless of the *label-preserving* augmentation. This understanding results in the graph in Figure 1 ($W$ is implicit) *excluding* $D$.

Von Kügelgen et al. (2021) showed that, under certain conditions, the above DGP is sound, and augmentation separates $C$ and $S$. However, the original picture misses an important variable: the domain $D$. By definition, observations are drawn from the distribution of the domain, thus latent variables $W$ are affected by the domain the data is generated from. Therefore it is unavoidable to incorporate a variable indicating domain $D$ in the figure. In sDG, $D$ is fixed in the sense that we are given just one domain. Due to the single source setting, we cannot distinguish what information is shared across different domains, leaving both $C$ and $S$ potentially affected by $D$. Hence, unless the discrepancy between the source and target is moderate, optimizing solely the augment-and-align objective (Eq. 1) would be insufficient to address the issue caused by a large domain gap.

**Learning to ignore**    To address a large domain shift, we begin with some observations. Conventional *augment and align* methods are vulnerable to domain shift in the sense that their effectiveness is affected by the augmentation's proximity to the domain shift. While advanced augmentation methods may simulate small shifts in distribution (e.g., MNIST $\rightarrow$ USPS in Digits), it is hard to approximate large domain shifts (e.g., PHOTO $\rightarrow$ SKETCH in PACS) (Section 5.1). If the gap between the source and target domain is large, failure in simulating domain shift would make its augment-invariant features less relevant to domain-invariant features, leading to overfitting to the source domain.

To avoid learning irrelevant features, we can think of a hypothetical regularizer that encourages the model to learn information relevant to domain-invariance, while discouraging domain-specific features. Certainly, this would require a condition that the regularizer be an oracle that can distinguish domain-invariant information. Using this oracle regularizer, we hope to solve the phenomena commonly associated with the large domain gap. Especially, the mid-training fluctuation of OOD performance, which was observed in earlier sDG works (Qiao et al., 2020; Li et al., 2021; Wang et al., 2021) but not discussed in-depth.[1] We view that the fluctuation is strongly correlated with the challenge in acquiring domain-invariant features under a large domain gap. We empirically observe that the level of domain gap between the *source* and *target* closely matches the magnitude of the mid-train fluctuation, where the increase in domain gap is simultaneously observed with the increase in fluctuation. Detailed information regarding the measure of domain gap is included in Section 5.2. We view mid-training fluctuation as a serious issue since it manifests that the simulated domains do not properly reflect unseen domains and, further, it harms the credibility of learned models due to uncertainty in their real-world performance. In the following section, we search for ways to implement the *hypothetical* oracle regularizer, inspired by works in knowledge distillation.

---

[1]On the contrary, the phenomenon has been discussed in the multi-DG literature (Arpit et al., 2022).

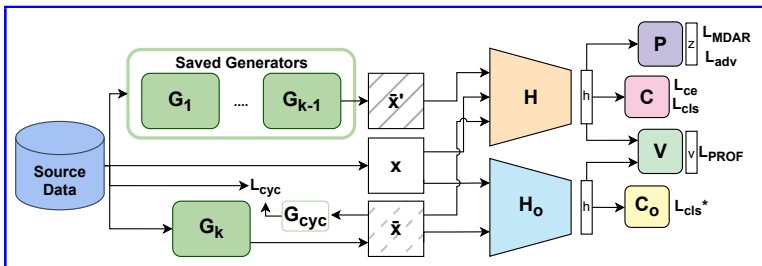

Figure 2: The illustration of our method. We sequentially train multiple generators $G_{1...K}$. The Oracle $H_o$ regulates the task model $H$'s learning process. During the training, multiple modules (e.g., $P, V, C$) are used for optimization.

## 4    LEVERAGING PRETRAINED MODELS TO LEARN DOMAIN INVARIANCE

We present a novel single source domain generalization method where the aim is to alleviate the issue of mid-train fluctuation. While the general principle of our approach is orthogonal to the type of the data, in this paper, we focus on image data. The overview of our architecture is depicted in Figure 2. At large, the architecture for our method involves three neural networks, a domain generator $G$, task model classifier $F$, and an oracle $O$. We sequentially learn multiple domain generators $\{G_k\}_{k=1}^{K}$ and use augmented samples created by the generators to train the task model $F$. More specifically, the generators provide the task model with *challenging* augmented samples, while the task model guides the generator to create *valid* augmentations. We train the above process using a combination of two losses: $L = L_f + w_g \cdot L_g$ where $L_f$ (Eq. 2) and $L_g$ (Eq. 10) are the loss used to train the task model and the generator, respectively, and $w_g \in \{0, 1\}$ controls the training of $G$.[2] The exact forms for $L_f$ and $L_g$ will become clear at the end of this section.

We build our method upon the idea that learning domain invariance solely from augmented domains is vulnerable to overfitting to the source, especially when the domain gap is too large to simulate via data augmentation. To alleviate this issue, we propose an oracle regularizer: under the hypothesis that the oracle is capable of generalizing well to unseen domains, we use the oracle to guide the task model to become less domain-dependent. Specifically, our oracle regularization objective regulates the sDG process via an alignment between the hidden feature representation of the task model and the oracle, which we name PROF. In the following section, we elaborate our ideas in depth.

**Notation**    We begin by introducing related notation regarding our method. To begin with, calligraphic letters are used to denote state space of a variable. For example, $\mathcal{X}$, $\mathcal{Y}$, and $\mathcal{H}$ respectively represent the space of the input image, intermediate feature representation, and labels.

- **Task model:** The task model $F = C \circ H$ consists of a feature-extractor $H : \mathcal{X} \to \mathcal{H}$ and a classification head $C : \mathcal{H} \to \mathcal{Y}$.

- **Generator:** A trainable generator $G : \mathcal{X} \to \mathcal{X}$ consists of an encoder-decoder architecture with a style-transfer module placed between the encoder and decoder.

- **Oracle:** The oracle model $O = C_o \circ H_o$ consists of a frozen feature-extractor $H_o : \mathcal{X} \to \mathcal{H}$ and a trainable classification head $C_o : \mathcal{H} \to \mathcal{Y}$. Task model $F$ and oracle model $O$ use separate feature-extractors ($H$ and $H_o$) to map the input data as intermediate representation and pass the representation to the classification head ($C$ and $C_o$) for the downstream classification task. For experimental purposes, we match the dimension of representation for the oracle and task model.

- **Distillation Head:** The distillation head $V : \mathcal{H} \to \mathcal{V}$ is used to impose regularization for the task model via oracle's representation. Instead of directly comparing the intermediate representation in $\mathcal{H}$, representations from $H_o$ and $H$ are mapped through the shared distillation head $V$, following the analysis of Gupta et al. (2022) on the efficacy of projection heads.

- **Projection Head:** Similar to the distillation head, the task model uses a projection head $P : \mathcal{H} \to \mathcal{Z}$ to project the intermediate representations into a different dimension. The projection

---

[2]Generally, $w_g = 1$ during the first half of the training epochs for $G_k$, then $w_g = 0$ to stop the training. This periodic control measure prevents the generator from deviating too far from the source (Li et al., 2021).

head is reserved for alignment of augmented views with MDAR, and its associated adversarial loss $L_{adv}$, thus not for PROF.

We train the task model $F$ using a weighted combination of multiple losses, namely, the cross-entropy classification loss of $x$ ($L_{ce}$) and $\bar{x}$ ($L_{cls}$; Equation (6)), with $L_{\text{PROF}}$ and $L_{\text{MDAR}}$ written as:

$$L_f = L_{ce}(C(H(x)), y) + L_{cls} + w_{\text{PROF}} \cdot L_{\text{PROF}} + w_{\text{MDAR}} \cdot L_{\text{MDAR}}, \tag{2}$$

where $w_{\text{PROF}}$ and $w_{\text{MDAR}}$ is a user-set parameter to activate two different methods, the oracle regularization PROF and our baseline MDAR. When training with the oracle regularizer (PROF) alone, $w_{\text{PROF}}$ is non-zero while $w_{\text{MDAR}}$ is set as $0$. Vice versa, $w_{\text{PROF}}$ is set as $0$ in our baseline (MDAR). We explain losses for PROF and MDAR in the next sections.

## 4.1 ORACLE REGULARIZER

We devise a novel learning method PROF (Progressive mutual information Regularization for Online distillation of Frozen oracles) to guide the learning process. PROF reformulates the sDG problem under the assumption that if there exists an oracle model $O$ that can generalize well to unseen domains, we can leverage the oracle to learn sDG. The objective for PROF can be formulated as:

$$L_{\text{PROF}}(x, \bar{x}, \lambda_{\text{PROF}}) = \sum_{x' \in \{x, \bar{x}\}} \text{BT}(V(H(x')), V(H_o(x')), \lambda_{\text{PROF}}) \tag{3}$$

where $x$ denotes the original sample and $\bar{x}$ the augmented view created by $G$, $\lambda_{\text{PROF}}$ is a user-set parameter, and Barlow Twins (BT) is defined as (Zbontar et al., 2021):

$$\text{BT}(z, z^+, \lambda) = \sum_i (1 - M_{ii})^2 + \lambda \sum_i \sum_{j \neq i} M_{ij}^2, \tag{4}$$

where $M$ refers to the cross-correlation matrix of the two positive-pair feature representations $z$, $z^+$, and $\lambda$ a user-set parameter.[3] BT (Eq. 4) is a feature-decorrelation loss originally introduced as a contrastive learning objective. BT is a combination of two terms balanced via a hyperparameter $\lambda$, where the first term $\sum_i (1 - M_{ii})^2$ aligns two representations by spurring the diagonal values in $M$ of $(z, z^+)$ to be 1 while the second term $\sum_i \sum_{j \neq i} M_{ij}^2$ minimizes redundancy in the representation by encouraging the off-diagonal values to be closer to $0$.

**Discussion on the Regularization via MI Optimization**    The idea of PROF is that we can distill the oracle's knowledge into the task model by maximizing the shared information between the two models. PROF aims to maximize the MI between the intermediate output features of the two feature-extractors $H$ and $H_o$. PROF functions as a regularization term that guides the task model from deviating too far from the oracle, encouraging the student task model to learn the oracle's behavior on data. From this perspective, an intended objective for PROF could be formulated as $\max_H I(H(x); H_o(x))$ where $I(X; Y) = \mathbb{E}_{p(x,y)}[\log p(x \mid y)/p(x)]$ indicates the mutual information (MI). However, directly estimating and optimizing MI are challenging, as the exact estimation of MI is intractable (Paninski, 2003). There exists InfoNCE loss (Oord et al., 2018) which adopts a lower bound of MI (Poole et al., 2019) as a surrogate objective for MI optimization:

$$I_{\text{NCE}}(X; Y) \triangleq \mathbb{E}\left[ K^{-1} \sum_{i=1}^{K} \log \frac{\exp(f(x_i, y_i))}{K^{-1} \sum_{j=1}^{K} \exp(f(x_i, y_i))} \right] \leq I(X; Y).$$

However, an issue of InfoNCE as a variational bound of MI is that InfoNCE requires a large batch size for convergence (Shrivastava et al., 2023; Hjelm et al., 2019), making it doubtful for use in small datasets (e.g., PACS). Consequently, we indirectly approximate InfoNCE with a feature decorrelation loss (Zbontar et al., 2021), based on empirical and theoretical results that show its functional proximity (Huang et al., 2021; Tao et al., 2022). Contrary to InfoNCE, the feature decorrelation converges effectively with small batch sizes and large vector dimensions.

Now we discuss the availability of an oracle. In reality, oracles may not be readily available. However, previous studies (Cha et al., 2022; Li et al., 2023) report that models pretrained from a large dataset or with deeper models tend to generalize better at unseen domains. Considering this, we utilize a model pretrained on a larger domain as an oracle. To preserve the knowledge of the oracle, we freeze the feature-extractor $H_o$ of the oracle.

---

[3]The actual computation involves a batch of data to obtain an empirical cross-correlation matrix.

### 4.2 Multi-Domain Alignment with Redundancy Reduction

We now introduce a novel alignment objective MDAR (Multi-Domain Alignment with Redundancy reduction) for sDG. MDAR aims to disentangle latent features that are invariant across multiple augmented views. We design MDAR as a fair baseline of the conventional *augment and align* method. In learning the $k$th generator $G_k$, we create an augmented view $\bar{x}$ for a batch of original samples $x$ using the $k$th generator $G_k$. We then randomly load two previously learned generators to construct two augmented views $\bar{x}'$ and $\bar{x}''$. With $\{x, \bar{x}, \bar{x}', \bar{x}''\}$, we encourage their representations vary in a similar way. Hence, we use BT (Eq. 4) over the representations for $\{x, \bar{x}, \bar{x}', \bar{x}''\}$ obtained through the projection head and feature extractor, $P \circ H$. That is, their cross-correlation matrix $M$ to be closer to an identity matrix. Our alignment loss $L_{\text{MDAR}}$ is written as:

$$L_{\text{MDAR}}(\mathbf{x} = \{x, \bar{x}, \bar{x}', \bar{x}''\}, \lambda_{\text{MDAR}}) = \sum_{x_i \neq x_j} \text{BT}(P(H(x_i)), P(H(x_j)), \lambda_{\text{MDAR}}), \qquad (5)$$

where $\lambda_{\text{MDAR}}$ a user-set parameter. Intuitively, via optimizing $L_{\text{MDAR}}$, we can train the task model in a way that multiple views (representations) are aligned. In terms of S-C disentanglement, MDAR encourages the retrieval of augment-invariant features. Different from the commonly used InfoNCE loss, our objective (Eq. 5) does not require negative pairs, thus works well on small batch sizes (Zbontar et al., 2021; Tsai et al., 2021), suitable for benchmarks like PACS.

In our conventional *augment and align* baseline experiment, we train our model with a variant of Eq. 2: $L_f = L_{ce}(C(H(x)), y) + L_{cls} + w_{\text{MDAR}} \cdot L_{\text{MDAR}}$.

### 4.3 Learnable Domain Shift Simulators

We sequentially train multiple generators to obtain varying simulated domains. The purpose of this process is to examine the behavior of models repeatedly trained on simulated domains, namely, the mid-train OOD fluctuation. To simulate domain shift, we must ensure that the augmented domain is label-preserved, while different from the source domain. Reflecting this, we adopt methods of Wang et al. (2021); Li et al. (2021) to assure the consistency of generated samples:

$$L_{cls}(\bar{x}, y) = L_{ce}(C(H(\bar{x})), y) + I(w_{\text{PROF}} > 0) \cdot L_{ce}(C_o(H_o(\bar{x})), y), \qquad (6)$$
$$L_{cyc}(x, \bar{x}) = \|x - G_{cyc}(\bar{x})\|_2, \qquad (7)$$

where $I$ is an indicator function. $L_{cls}$ is a cross-entropy loss that assures the validity of the generated samples $\bar{x}$ based on predictions from task model $F$ (also from oracle $O$ if PROF is employed.) $L_{cyc}$ ensures that the output of $G$, can be recovered to the original input image when passed through the inversed generator $G_{cyc}$ (Zhu et al., 2017).

Next, we encourage the generator to create diverse augmentations with the following objectives:

$$L_{div}(\bar{x}_1, \bar{x}_2) = -\|\bar{x}_1 - \bar{x}_2\|_2, \qquad (8)$$
$$L_{adv}(x, \bar{x}, \lambda_{adv}) = -\text{BT}(P(H(x)), P(H(\bar{x})), \lambda_{adv}). \qquad (9)$$

$L_{div}$ is a negated L2-norm between two augmented views ($\bar{x}_1, \bar{x}_2$) of a batch $x$ created with the generator. Intuitively, optimizing with $L_{div}$ encourages the generator to augment diverse samples, preventing collapse. $L_{adv}$ is an adversarial loss function designed to reverse the alignment process by negating the feature-decorrelation loss used in Eq. 4.

We train the generator with the weighted sum $L_g$ of the above four objectives:

$$L_g = L_{cls} + w_{cyc} \cdot L_{cyc} + w_{div} \cdot L_{div} + I(w_{\text{MDAR}} > 0) \cdot w_{adv} \cdot L_{adv}, \qquad (10)$$

where $L_{adv}$ is active only if MDAR is used.

## 5 Experiment

We first present our experimental settings including datasets and architectures. Then, we report experimental results using the accuracy for each target domain, as well as the mean accuracy over all target domains. We designed our experiments to be reproducible.

## 5.1 Experimental Settings

**Datasets** Following the experimental settings in prior sDG works (Qiao et al., 2020; Li et al., 2021; Wang et al., 2021), we adopted three broadly used benchmarks for our sDG problem, along with an additional benchmark. **PACS** (Li et al., 2017) is widely used to test the generalizability of trained models against domain shift. It consists of 4 domains of differing styles (Photo, Art, Cartoon, and Sketch) with 7 classes. In default, we train our model with the Photo domain and evaluate the remaining target domains. We also present additional experiments in Appendix A.1. Among the selected benchmarks, PACS is the main target of PROF due to its large gap between domains. **Corrupted CIFAR-10** (i.e. CIFAR-10-C) is a benchmark to test the image classifier robustness under distortion (Hendrycks & Dietterich, 2019). We train our model with the train split of the CIFAR-10 (Krizhevsky & Hinton, 2009) dataset and test the model accuracy in CIFAR-10-C. We evaluate the robustness of the model with 19 types and 5 levels of corruption. Unlike other benchmarks, we expect that the CIFAR-10-C is sufficient with conventional *augment and align* methods, as each target domain is created via augmentation of the source domain. **Digits** dataset is a popular benchmark for sDG, comprised of 5 different digit classification datasets, MNIST (Deng, 2012), SVHN (Netzer et al., 2011), MNIST-M (Ganin et al., 2015), SYNDIGIT (Ganin & Lempitsky, 2015), USPS (Le Cun et al., 1989). In our experiment, we train our model with the first 10,000 samples of the MNIST dataset and assess its generalization accuracy across the remaining four domains. **Office-Home** dataset (Venkateswara et al., 2017) is a common benchmark for DG, but not used for sDG. The benchmark consists of 4 datasets (Real-world, Art, Clipart, Product) with differing styles with 65 classes. We train our model on the Real-world domain and evaluate the remaining domains.

**Implementation** In all experiments, we utilized the identical network architectures used in previous sDG works. For PACS, we adopted AlexNet (Krizhevsky et al., 2012) pretrained on Imagenet (Russakovsky et al., 2014). For corrupted CIFAR-10, we used a Wide Residual Network (Zagoruyko & Komodakis, 2016) of depth 16, and width 4. For Digits, we used the identical network architecture (i.e. conv-pool-conv-pool-fc-fc-softmax) used in previous works. For Office-Home, we used a ResNet18 (He et al., 2016) pretrained on ImageNet-1K dataset (Russakovsky et al., 2014). For an oracle, we selected pretrained models appropriate for each experiment. For PACS and Office-Home, we chose a RegNetY-16GF (Radosavovic et al., 2020) pretrained on Instagram dataset with SWAG (Supervised Weakly through hashtAGs) (Singh et al., 2022) following experimental reports of Cha et al. (2022); Li et al. (2023). For Corrupted CIFAR-10, we selected an Imagenet pretrained ResNet50. For Digits, we followed the practice of Cha et al. (2022) and used a true oracle pretrained on both the source and target domains of Digits. All oracles are finetuned on the source domain (e.g. Photo, CIFAR-10, MNIST, Real World) and frozen. We test the sensitivity of the hyperparameters using the validation split of the source dataset. Details regarding the training hyperparameters, pretraining process, training process, and the generator module are reported in Appendix B.4, Appendix B.3, Appendix B.2, and Appendix B.1, respectively.

## 5.2 Experimental Results and Analysis

Here we present experimental results over the four benchmark datasets, examination on domain gaps and the effect of PROF.

**Experiment with PACS** The aim of the PACS experiment is to show that PROF functions as a stable regularizer for sDG, reducing the mid-train OOD fluctuation reported in conventional *augment and align* methods. The results of the PACS experiment are reported in Table 1 where AN, RN, M, and P stands for AlexNet, ResNet18, MDAR, and PROF, respectively.

First, we compare the generalization accuracy. Training AlexNet with PROF (Eq.(2)) showed results close to the current SOTA (Wan et al., 2022) without the use of alignment. Furthermore, we showed state-of-the-art performance in the Sketch domain, where domain gap is considered to be the largest. Similarly, our *augment and align* baseline using

Table 1: sDG accuracy on PACS.

| Method | A | C | S | Avg. |
|---|---|---|---|---|
| ERM [30] | 54.43 | 42.74 | 42.02 | 46.39 |
| JiGen [5] | 54.98 | 42.62 | 40.62 | 46.07 |
| RSC [23] | 56.26 | 39.59 | 47.13 | 47.66 |
| ADA [10] | 58.72 | 45.58 | 48.26 | 50.85 |
| ME-ADA [74] | 58.96 | **51.05** | 58.42 | 51.00 |
| L2D (AN) [67] | 56.26 | 51.04 | 58.42 | 55.24 |
| MetaCNN [65] | 54.05 | 53.58 | 63.88 | 57.17 |
| Ours (AN+P) | 52.46 | 50.29 | **66.79** | 56.52 |
| Ours (AN+M) | 57.54 | 46.89 | 64.93 | 56.45 |
| Ours (AN+MP) | **58.96** | 45.86 | 64.57 | 56.46 |
| L2D (RN) | 68.41 | 43.56 | 48.84 | 53.60 |
| L2D (RN+M) | 57.57 | **50.09** | 65.51 | **57.72** |
| Ours (RN+M) | 58.25 | 47.35 | **67.81** | 57.80 |
| Ours (RN+P) | 58.42 | 48.29 | 66.68 | 57.80 |
| Ours (RN+MP) | 64.06 | 42.06 | **73.98** | 60.03 |

MDAR also showed an accuracy close to SOTA. However, we observe that the method using MDAR displays a fluctuation of OOD performance after a certain point (i.e. $K > 5$). The behavior worsened as training continued. On the contrary, training with PROF resulted in stabilization of the OOD performance, mitigating fluctuations, quantified as the reduction in variance across the target domain accuracy in $K > 5$ (Art: 3.39→1.27, Cartoon: 5.22→2.49, Sketch: 7.23→5.30). The mid-train OOD stabilization effect is depicted in Figure 3. Finally, we show the competitiveness of our baseline (MDAR). We applied MDAR to an existing sDG method (Wang et al., 2021) by replacing the InfoNCE loss with MDAR. We observe a wide improvement over the conventional methods under certain conditions, as recorded in the last rows of Table 1.

**Experiment with Corrupted CIFAR-10**   We present results over CIFAR-10-C (Table 2) where we compare the effectiveness of the conventional *augment and align* method (MDAR) and PROF under *small* domain shifts. We report the average accuracy (%) of each corruption category (Weather, Blur, Noise, Digits), and the average accuracy of all categories. Our baseline using MDAR marked scores close to the current SOTA (Wan et al., 2022) in two categories Weather and Blur while falling behind in others, Noise and Digital. We report that the OOD performance of the CIFAR-10-C is greatly affected by the design of the domain simulator $G$. On the

Table 2: sDG accuracy on Corrupted CIFAR-10.

| Method | W | B | N | D | Avg. |
|---|---|---|---|---|---|
| ERM [30] | 67.28 | 56.73 | 30.02 | 62.30 | 54.08 |
| CCSA [42] | 67.66 | 57.81 | 28.73 | 61.96 | 54.04 |
| d-SNE [71] | 67.90 | 56.59 | 33.97 | 61.83 | 55.07 |
| M-ADA [50] | 75.54 | 63.76 | 54.21 | 65.10 | 64.65 |
| L2D [67] | 75.98 | 69.16 | 73.29 | 72.02 | 72.61 |
| MetaCNN [65] | 77.44 | 76.80 | **78.23** | **81.26** | **78.45** |
| Ours M | **77.10** | **76.35** | 67.94 | 76.57 | 74.49 |
| Ours P | 72.61 | 70.30 | 54.26 | 71.97 | 67.28 |

contrary, our method using PROF marked results lower than our baseline MDAR. This is anticipated as we view the domain gap to be small between different datasets in the CIFAR-10-C, whereas PROF is designed for use under large domain discrepancies.

**Experiment with Digits**   We share our results on the digit experiment on Table 3. The aim of the Digits experiment is to validate the efficacy of the oracle regularization (PROF) and present the strength of our baseline (MDAR). We underline in advance that in the Digits benchmark, we could not obtain a pretrained model fit for use as the oracle. Hence, we follow the practice of Cha et al. (2022) and use a *true* oracle, a model pretrained on both the source and target domains. Our method with PROF showed a large drop in mid-train

Table 3: sDG accuracy on Digits.

| Method | SVHN | M-M | S-D | USPS | Avg. |
|---|---|---|---|---|---|
| ERM [30] | 27.83 | 52.72 | 39.65 | 76.94 | 49.29 |
| JiGen [5] | 33.80 | 57.80 | 43.79 | 77.15 | 53.14 |
| M-ADA [50] | 42.55 | 67.94 | 48.95 | 78.53 | 59.49 |
| L2D [67] | 62.86 | 87.30 | 63.72 | 83.97 | 74.46 |
| PDEN [36] | 62.21 | 82.20 | 69.39 | 85.26 | 74.77 |
| MetaCNN [65] | 66.50 | **88.27** | 70.66 | **89.64** | 78.76 |
| Ours M | **68.29** | 81.88 | 76.24 | 88.79 | **78.80** |
| Ours P | **74.50** | 87.98 | **78.67** | 86.15 | **81.82** |

OOD fluctuation compared to the baseline (M-M: $2.56 \to 1.17$, USPS: $3.48 \to 1.11$, SVHN: $3.58 \to 1.95$, S-D: $2.36 \to 2.10$). The OOD stabilization effect is illustrated in Figure 5 (Appendix A.2). Furthermore, PROF displays superior generalization accuracy (81.82) compared to existing methods, which is expectable from the perspective of knowledge distillation.[4] Similarly, our baseline using MDAR surpassed state-of-the-art records. Analysis on PROF and MDAR continue in Appendix A.2.

**Experiment with Office-Home**   The aim of the Office-Home experiment is to stress the effectiveness of PROF for mitigating the issues of stochasticity under large distributional shifts. We report the results of the Office-Home experiment on Table 4, where RN stands for ResNet18.

In terms of performance, our method using PROF displayed a strong advantage over the conventional baseline with MDAR. In terms of OOD fluctuation, regularizing with PROF displayed a stabilization of the OOD performance, measured as the reduction in variance across the target domain accuracy (Art: $10.63 \to 8.23$, Clipart: $2.17 \to 2.05$, Product: $7.46 \to 6.41$). The stabilization effect is illustrated in Figure 6 (Appendix A.3). Detailed analysis on the Office-Home experiment is reported in Appendix A.3.

Table 4: sDG accuracy on Office-Home.

| Method | Art | Clipart | Product | Avg. |
|---|---|---|---|---|
| ERM (RN) | 52.78 | 40.19 | 68.73 | 53.90 |
| Ours (RN +M) | 53.39 | 43.38 | 66.25 | 54.34 |
| Ours (RN +P) | **55.25** | **46.69** | **69.26** | **57.07** |

**Experiment on domain gaps**   We show results that display a strong correlation between the level of domain gap and the magnitude of mid-train fluctuation. In Digits, it is commonly viewed that

---

[4]As we use the *true* oracle, we do not claim state-of-the-art for PROF.

the gap between the source (MNIST) and the target is greater in certain datasets (e.g., SVHN and SYNDIGIT) over others (e.g., MNIST-M and USPS). For instance, the baseline OOD accuracy is much higher in some target domains as opposed to others, in the order of: USPS(76.94%) > MNIST-M(52.72%) > SYNDIGIT(39.65%) > SVHN(27.83%), as recorded in Table 3. We elaborate the domain gap further in Appendix C. Interestingly, in our baseline experiment using the conventional *augment and align* method, we find that the mid-train fluctuation follows the same order: USPS(1.211) < MNIST-M(1.1795) < SYNDIGIT(4.938) < SVHN(5.106), measured by the variance of the OOD accuracy after $K > 5$. A similar pattern is observed on PACS (Table 1), where the baseline OOD accuracy order Art (54.43%), Cartoon (42.74%), and Sketch (42.02%) matches the order of the mid-train fluctuation: Art (3.39), Cartoon (5.22), and Sketch (7.23). We view that these results empirically support the correlation between domain gap and mid-train fluctuation.

**Effect of PROF**  We study further the effect of PROF on OOD generalization. Experimental results are illustrated in Figure 3 (A, C, and S are from PACS and M and P from MDAR and PROF.) The stabilization effect of PROF is repeatedly confirmed across many benchmarks including Digits (Figure 5) and Office-Home (Figure 6). We view that the reduction in mid-train OOD fluctuation ultimately increases the credibility of the model at test time. In real-world settings, a model with large fluctuation is unreliable since its performance may drop unknowingly. Hence, a reduction in fluctuation is closely synonymous with model consistency.

Figure 3: OOD accuracy (%) on PACS (Source: Photo)

Conversely, using PROF showed limited impact in enhancing generalization accuracy. In experiments performed with AlexNet, the increase in OOD accuracy was not significant (Table 1). However, using the ResNet18 architecture, OOD accuracy on both Art and Sketch domains benefited from using PROF. Similarly in the Office-Home dataset, using PROF with ResNet18 largely increased the accuracy (Table 4). Our notion is that the model architecture (e.g., width and depth) affects the knowledge transfer capability, though further research is required.

**Study of Hyperparameters**  We further present an examination of our method's hyperparameters. We empirically observe that our method is resilient to individual changes in hyperparameters. The details of the analysis are reported in Appendix A.5.

## 6 CONCLUSION

This paper presents PROF (Progressive mutual information Regularization for Online distillation of Frozen oracles), a novel oracle regularizer to address single source domain generalization under a large domain discrepancy. We underscore the vulnerability of learning robustness via augmentation, which is observed as large fluctuations in the OOD performance during the training process. To mitigate this issue, PROF leverages pretrained oracles to guide the model to learn features that are less domain-specific, via maximization of the feature-level mutual information between the learning model and the oracle. Experiments on multiple datasets (PACS, Digits, Office-Home) demonstrate that PROF can stabilize the fluctuations associated with large domain gaps. We further introduce a strong baseline method with MDAR (Multi-Domain Alignment with Redundancy Reduction) for a fair comparison with PROF. Training with MDAR showed state-of-the-art performance in Digits and displayed a boost in performance when applied to existing methods.

## ACKNOWLEDGEMENT

This work was partly supported by IITP (2022-0-00953-PICA/50%) and NRF (RS-2023-00211904/50%) grant funded by the Korean government (MSIT).

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

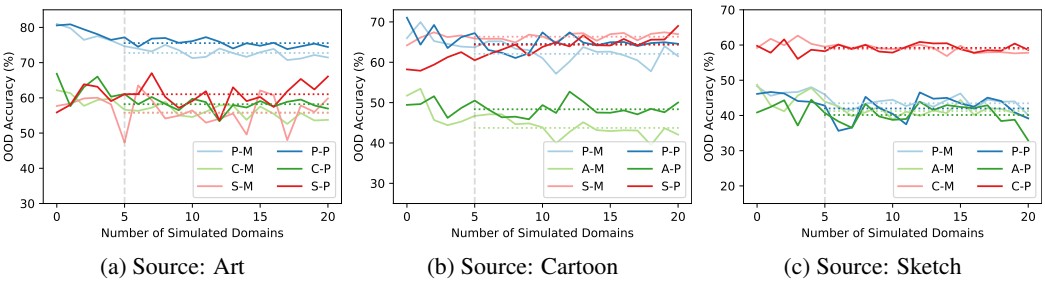

(a) Source: Art  (b) Source: Cartoon  (c) Source: Sketch

Figure 4: OOD accuracy (%) on PACS (Additional)

# A    EXPERIMENTAL RESULTS

## A.1    EXPERIMENTS ON PACS (CONTINUED)

Here we present the results of additional experiments with the PACS benchmark.

Previous experiments on the PACS benchmark only used the Photo dataset as the source domain. In the following section, we report other cases where the source domain is changed (e.g., Art, Cartoon, Sketch). Here, we will denote each experiment as *Art as source*, *Cartoon as source*, and *Sketch as source*, respectively.

In Table 5, we report the sDG accuracy of our two methods, MDAR and PROF, where AN, M, and P stands for AlexNet, MDAR, and PROF, respectively. Each row in the table displays the source domain, backbone type, and the training method (M/P). In cases where Art or Cartoon is used as source domain, training with our oracle regular-

Table 5: sDG accuracy on PACS (Full).

| Method | P | A | C | S | Avg. |
|---|---|---|---|---|---|
| | | Source: Photo | | | |
| Ours (AN+P) | ——— | 52.46 | 50.29 | 66.79 | **56.52** |
| Ours (AN+M) | ——— | 57.54 | 46.89 | 64.93 | 56.45 |
| | | Source: Art | | | |
| Ours (AN+P) | **78.07** | ——— | **66.04** | **63.15** | **69.09** |
| Ours (AN+M) | 77.53 | ——— | 59.39 | 60.04 | 65.65 |
| | | Source: Cartoon | | | |
| Ours (AN+P) | 64.57 | **50.02** | ——— | **69.00** | **62.04** |
| Ours (AN+M) | 65.20 | 47.10 | ——— | 65.81 | 59.37 |
| | | Source: Sketch | | | |
| Ours (AN+P) | 46.25 | 44.31 | 61.60 | ——— | 50.72 |
| Ours (AN+M) | **48.03** | **47.83** | 60.32 | ——— | **52.06** |

ization PROF marked higher OOD accuracy then its counterpart. On the other hand, PROF suffered when Sketch was set as the source domain, falling behind the baseline MDAR. Our hypothesis is that this behavior is triggered by the subpar performance of the oracle. To elaborate, the oracle used on the *Sketch as source* experiment displayed low OOD accuracy on the target domains, unsuitable for effective oracle regularization (Photo: 51.61%, Art: 39.39%, Cartoon: 56.85%).

Next, we present the analysis on mid-train OOD fluctuation in each experimental configuration. When the source domain is set as Art, employing PROF resulted in yielded a stabilization of the OOD performance, effectively mitigating fluctuations. The fluctuation was quantified as the reduction in variance across the target domain accuracy in $K > 5$. When compared with the conventional *augment & align* method MDAR, our regularization method PROF displayed large reductions in variance (Photo: 1.71→1.17, Cartoon: 3.13→2.97, Sketch: 21.50→11.22). The mid-train OOD fluctuation when source is set as Art, is depicted in Figure 4a.

Similarly, when the source domain is configured as Cartoon, PROF displays similar stabilization of the mid-train OOD performance. Using PROF allows a reduction in fluctuation, measured as variance (Photo: 5.15 → 3.06, Art: 5.00 → 3.07, Sketch: 0.70 → 3.91). We note that the stabilization effect in Sketch is relatively lower than that of other target domains, even lower than our *augment & align* baseline MDAR. The mid-train fluctuation is demonstrated in Figure 4b.

Lastly, we report the experimental results where the source was set as Sketch. In the *Sketch as source* experiment, we observe that PROF not only suffers in terms of performance but also exhibits instability. PROF displayed high variance in mid-train performance when compared to the baseline (Photo: 2.46 → 10.41, Art: 2.33 → 7.99, Cartoon: 1.01 → 1.04). The fluctuation is illustrated in Figure 4c. While a clear explanation is absent, we view that this phenomenon is caused by the under-performance of the oracle in the *Sketch as source* experiment. This result displays a clear example of the problems associated with the obstacles regarding the oracle, where obtaining an oracle may not be readily available. We further discuss the issue with oracles in the following section, Appendix D

## A.2 EXPERIMENTAL RESULTS ON DIGITS (CONTINUED)

Here we continue our analysis on the results of the Digits Experiment. In Section 5, we demonstrated that our regularization method PROF successfully mitigates issues of OOD fluctuation, measured as variance. This is illustrated in Figure 5 (M and P are from MDAR and PROF.). One notable observation is the significant increase in OOD generalization accuracy (81.82) when using PROF, in Table 3. As mentioned in the footnote, we do not claim this score to be state-of-the-art, as the true oracle is used. From the perspective of knowledge distillation, this is anticipated as the true oracle is already generalized to the target domains. In comparison, the approximated oracle in PACS does not guarantee robustness in the target domains, despite its higher generalizabil-

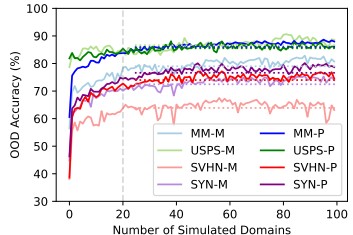

Figure 5: OOD accuracy (%) on Digits

ity. This confirms that a gap between the approximated oracle and the true oracle exists, which is a limitation that we acknowledge. We provide further analysis on the oracle in Appendix D

Next, we discuss the results of our baseline experiment using MDAR. As mentioned in the main paper, our baseline surpassed state-of-the-art in Digits. In SVHN and SYNDIGIT (S-D), we show large improvement, while results in MNIST-M (M-M) show slight deficiency. Similar to existing methods, we refrain from using any form of manual data augmentation. We find that in Digits, increasing the number of simulated domains ($K$) helps OOD generalization. Both our baseline (MDAR) and PROF benefited from long training ($K > 100$).

## A.3 EXPERIMENTAL RESULTS ON OFFICE-HOME (CONTINUED)

Here we continue our analysis of the results of the Office-Home Experiment. The Office-Home benchmark is not commonly used in the sDG literature, but we include the benchmark to bring attention to an important question: Is augmentation reliable for sDG?

As described in Table 4, augmentation-based approaches do show a boost in OOD accuracy. However, the effect gradually disappears with a sharp decline in OOD accuracy, as depicted in Figure 6. (A, C, and P are abbreviations of Art, Clipart, and Product domains, while M and P are from MDAR and PROF.) This downward trend is also spotted on other benchmarks, but not as intense. We believe that this phenomenon aligns with our analysis of the uncertainty of utilizing augmentation for OOD generaliza-

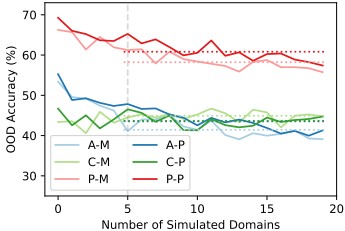

Figure 6: OOD accuracy (%) on Office-Home

tion. Our hypothesis is that the distributional gap within the Office-Home benchmark may be more intense than conventional sDG benchmarks (e.g., Digits, Corrupted CIFAR-10, PACS). The phenomenon brings novel questions on the efficacy of augmentation-based generalization methods. We believe that further research is required. Nonetheless, even in this case, PROF continues to stabilize the learning process, showing a smaller variance than our baseline (MDAR).

## A.4 A SYNERGISTIC APPROACH: COMBINED USE OF MDAR AND PROF

In this section, we report the effect of using MDAR and PROF simultaneously. While PROF was designed for use without an alignment term (e.g., MDAR), we tested the effect of combining the two terms together. We observe that the synergistic method of PROF and MDAR triggered some differences in the training process.

Regarding the OOD accuracy, the synergistic method marked Art: 58.96%, Cartoon: 45.86%, Sketch: 64.57%, an average of 56.46% with AlexNet, as seen in Table 1. While the accuracy is slightly higher than using MDAR alone (56.45%), we view that the synergistic method does not significantly benefit the OOD performance. On the other hand, applying the synergistic method with a ResNet18 backbone showed a rise in OOD accuracy by a large

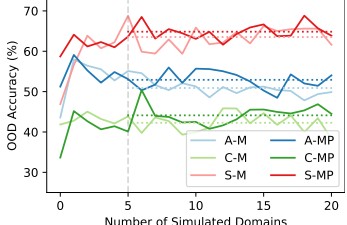

Figure 7: OOD accuracy (%) on PACS (MDAR + PROF)

gap 1. Further research is necessary to provide an understanding of this behavior as no definitive explanation currently exists, while our hypothesis is that the model architecture may have caused the phenomenon.

Regarding the mid-train OOD fluctuation, the synergistic method was not able to reduce fluctuations across Art and Cartoon, while reducing the fluctuation in Sketch. (Art: 3.39→4.50, Cartoon: 5.22→5.86, Sketch: 7.23→3.52) Similar to previous experiments, the mid-train OOD fluctuation was quantified with the variance across the target domain accuracy in $K > 5$. The mid-train OOD fluctuation is depicted in Figure 7 (A, C, and S are from PACS and M and MP from MDAR and MDAR+PROF, the synergistic method.). Our hypothesis is that the two terms may have disrupted each other, while a clear explanation for this phenomenon remains elusive. We believe that additional research is needed to produce an effective synergy of both methods.

### A.5   STUDY OF HYPERPARAMETERS (CONTINUED)

We explore our method's sensitivity to hyperparameters. ($\lambda_{\text{PROF}}$): $\lambda_{\text{PROF}}$ is the hyperparameter used for PROF that operates as the balancing weight of the two functions in Equation (4). We begin with the value in the original paper of Zbontar et al. (2021) with $\lambda_{\text{PROF}} = 0.005$, and an alternate value $\frac{1}{d}$ introduced in Tsai et al. (2021) where $d$ is the length of a vector in $\mathcal{V}$ (distillation head output space). We observe that our method is resilient to the switch between two candidate values of $\lambda_{\text{PROF}}$ although we cannot guarantee they are optimal. ($\lambda_{\text{MDAR}}$ and $\lambda_{adv}$): The study on $\lambda_{\text{MDAR}}$ and $\lambda_{adv}$ is processed similar to $\lambda_{\text{PROF}}$. Switching between $\lambda = 0.005$ and $\frac{1}{p}$ posed no notable impact on the learning process, where $p$ is the length of a vector in $\mathcal{P}$ (projection head output space). While we cannot guarantee an optimal value. ($w_{adv}$, $w_{cyc}$, $w_{div}$): We optimize the hyperparameters $w_{adv}$, $w_{cyc}$, $w_{div}$ using grid search. We find that as long as the weight-multiplied loss ($wL$) is situated on the $(0, 1)$ range, there is no significant impact on performance.

## B   IMPLEMENTATION DETAIL

In this section, we report the implementation details of our method.

### B.1   MODEL ARCHITECTURE

We report the details of model architectures used in our experiments. All models were built to match the architecture used in previous studies.

**Task Model**   The task model architecture varies in each experiment. For each experiment, we report the feature extractor $H$, including an additional layer (i.e. buffer) used to match the feature extractor's output dimension to the oracle's.

The model used in the PACS experiment is AlexNet (Krizhevsky et al., 2012). The model consists of 5 convolutional layers with channels of $\{96, 256, 384, 384, 256\}$, followed by two fully-connected layers of size $4096$ units. The buffer is a 2-layered MLP that maps the output dimension $4096$ to that of the oracle (RegNetY-16GF), which is $3024$. Hence, the final output dimension of the feature extractor is $3024$.

The model used in the Corrupted CIFAR-10 experiment is a Wide Residual Network (i.e. WRN) of width $w = 4$ and depth 16 (Zagoruyko & Komodakis, 2016). WRN is a model that boosts its performance by widening the network by a certain factor $w$. The model consists of 4 network blocks with channels incrementally increasing as $\{16, 16w, 32w, 64w\}$. Specifically, the 4 blocks refer to an initial convolutional layer, followed by three additional network blocks. We further follow the original WRN design and set the dropout rate as $0.3$. The buffer is a 2-layered MLP that maps the output dimension $256$ to that of the oracle (ResNet50), which is $2048$. Hence, the final output dimension of the feature extractor is $2048$.

For the model used in the Digits experiment, please refer to Section 5.1. The architecture consists of two $5 \times 5$ convolutional layers, with 64 and 128 channels respectively. Each convolutional layer is followed by a MaxPooling layer ($2 \times 2$). The network also includes two fully connected layers with sizes of $1024$, $1024$ being the final output dimension of the feature extractor. Since we do not employ oracle for the Digits experiment, a buffer was not added.

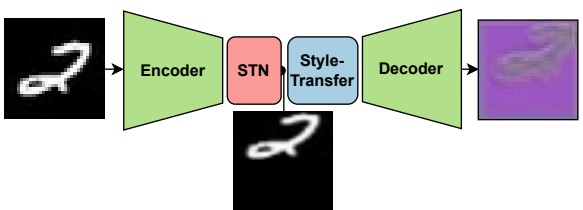

Figure 8: The illustration of the Generator.

**Generator**  In this section, we describe the generator in detail. While the design of the generator slightly varies in each experiment, the basic architecture is the same. The generator consists of an encoder and a decoder, with a spatial transformer network (STN) and a style-transfer module in between the encoder and the decoder. The four components are placed in the order of `Encoder – STN – Style-Transfer – Decoder`.

We begin by illustrating the overall process of how an image is augmented by the generator. First, the input image is passed through the encoder to get a feature representation vector. The feature vector is then passed through the STN and the style-transfer module for modification. The modified vector is then reconstructed via a decoder, returning an augmented image. The mentioned process is illustrated in Figure 8. In the figure, we depict how each module modifies the input image.

STN is a module that learns to perform spatial transformations on the input (Jaderberg et al., 2015). During the process, the STN module learns transformation parameters, where the parameters each define the magnitude of spatial transformations (e.g., rotation, scaling, translation). The STN module can be inserted at any point in the generator, allowing the generator to selectively transform the data up to a degree that is label-preserving. We place the STN right after the Encoder, following the experimental results of the original paper (Jaderberg et al., 2015). In Figure 8, we can see that the STN performs spatial transformations, creating the modified image at the middle. An advantage of STN is that no additional requirements are needed for training the module.

The style-transfer module modifies the features of the input image by adjusting the mean and standard deviation of the image features. This is performed using a normalization technique called Batch-Instance Normalization (i.e. BIN) (Nam & Kim, 2018). BIN selectively normalizes the features of the input image that are of less significance, while preserving features that are important. Note that this module is a modified version of the AdaIN method introduced in Huang & Belongie (2017), where we switched the normalization method from Instance Normalization (Ulyanov et al., 2016) to BIN for effective style transfer.

We share the results of applying these modifications in Figure 9. Whilst previous augmentation methods (Li et al., 2021; Wang et al., 2021) were limited to manipulating certain attributes (e.g., color, stroke), our method further allows spatial manipulations (e.g., shape, location). For instance, in the right image of Figure 9, we can observe that the images generated using our method displayed a large variance in shape, position, and color. This modification is inspired by recent studies on domain shift (Kaur et al., 2022; Wiles et al., 2021), which revealed that domain shift occurs on a variety of levels. However, an observable limitation is that the STN cannot transform complex images as in PACS, as small spatial modifications vastly change the semantics of the image. As depicted in Figure 10, the effect of the spatial modification is limited on PACS images.

**Oracle**  Here, we report the architecture of the oracle. The oracle varies on the type of the experiment, (1) a RegNetY-16GF for the PACS and Office-Home experiment, (2) a ResNet50 for the corrupted CIFAR-10 experiment.

The RegNetY-16GF is a variant of the RegNet family, a line of models introduced in (Radosavovic et al., 2020) for image classification. The name of the model indicates its configurations, where the "Y" indicates the convolution method, and the "16GF" represents the model's capacity or complexity. We implement the model, and its model weights using the torchvision (Falbel, 2023) library. We used the weights pretrained via end-to-end fine-tuning of the original SWAG (Singh et al., 2022) weights on the ImageNet-1K data (Russakovsky et al., 2014). We then fine-tuned the pretrained model again with the Photo domain of PACS for 200 epochs, with a learning rate of $1e-4$ using

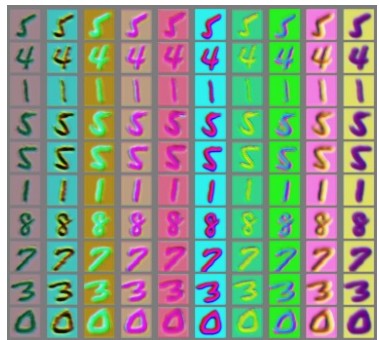
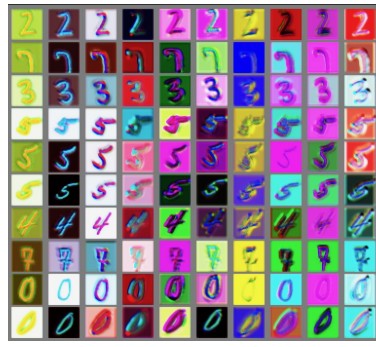

(a) Conventional Style-transfer  (b) STN + Style-transfer

Figure 9: The illustrated comparison of the generators.

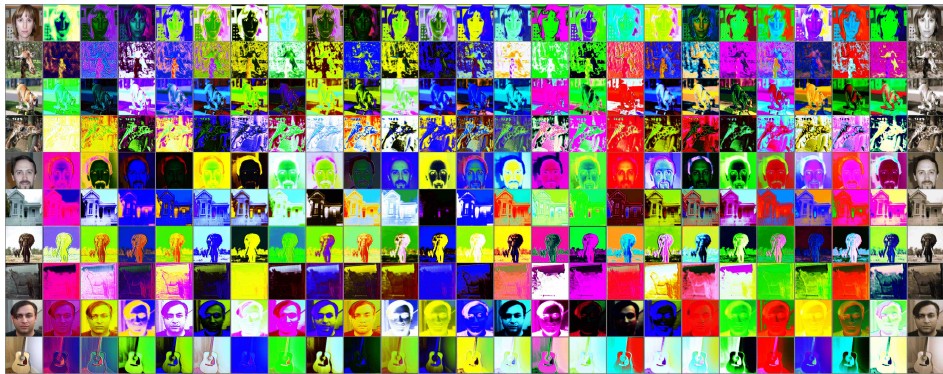

Figure 10: The illustration of generated images (PACS).

the SGD optimizer and the Cosine Annealing learning rate scheduler, a batch size of 64. For the Office-Home, we fine-tuned the pretrained model with the Real World domain of Office-Home for 30 epochs, using the SGD optimizer and the Cosine Annealing scheduler, a batch size of 16.

The ResNet50 is a variant of the ResNet family, a series of image classification models introduced in He et al. (2016). The name of the model indicates its depth, where "50" marks the number of layers. We implemented the model and its model weights using the torchvision library. For ResNet50, we used the weights pretrained with the ImageNet-1K dataset. We finetuned the pretrained ResNet50 with the CIFAR-10 dataset, the source domain of the corrupted CIFAR-10 experiment. In detail, we trained for 100 fine-tuning epochs, with a learning rate of $1e-4$ with the SDG optimizer and the Cosine Annealing learning rate scheduler, a batch size of 64.

### B.2  MODEL TRAINING

In this section, we elaborate on the details of the training process. We explicitly state the training hyperparameters (e.g., number of simulated domains ($K$), number of inner training loops for each generator, learning rate, the type of the optimizer, learning rate scheduler, and batch size). We further state the configurations of the projection heads (e.g., projection dimension ($\mathcal{Z}$) of the projection head $P$, projection dimension ($\mathcal{D}$) of the distillation head $V$).

**PACS**  For the PACS experiment, we set $K$ as 20, training each generator with 30 inner loops. During the first 15 inner loops we train the generator, and stop the training during the last 15 loops. We manually set the number of epochs by analyzing the training behavior of the generators. We set the learning rate as $1e-4$, using the Adam optimizer (Kingma & Ba, 2015). The batch size was set as 64. Regarding the model architecture, both the projection dimension ($\mathcal{Z}$) and the distillation head projection dimension ($\mathcal{V}$) were set as 1024.

**Corrupted CIFAR-10**  For the Corrupted CIFAR-10 experiment, we set K as 20, and 20 inner loops. During half (10) of the inner loops, we trained the generator and stopped the training during the remaining 10 inner loops. We set the learning rate as $1e-4$, with the Adam optimizer. The batch size was set as 256. The projection dimension ($\mathcal{Z}$) and the distillation head projection dimension ($\mathcal{V}$) were both set as 512.

**Digits**  For the Digits experiment, we set $K$ as 100, with 10 inner loops. Similar to the above two experiments, we trained the generator for 5 epochs and stopped the training for the other 5. Furthermore, the learning rate was tuned as $1e-4$, using the Adam optimizer. The batch size was set as 128. Finally, both the projection dimension ($\mathcal{Z}$) and the distillation head projection dimension ($\mathcal{V}$) were as 128.

**Office-Home**  For the Office-Home experiment, we set $K$ as 20, training each generator with 30 inner loops. During the first 15 inner loops we train the generator, and halted training for the remaining 15 loops. Similar to other cases, we set the number of epochs by analyzing the training behavior of the generators. The learning rate was set as $1e-4$, using the Adam optimizer. The batch size was set as 64. Regarding the model architecture, both the projection dimension ($\mathcal{Z}$) and the distillation head projection dimension ($\mathcal{V}$) were set as 512.

## B.3 MODEL PRETRAINING

In this section, we report the information regarding the pretraining process. As mentioned above, we pretrained our task model with the source domain prior to the main training procedure. We announce the number of pretraining epochs, the learning rate, the optimizer, the learning rate scheduler, and the batch size.

**PACS**  We pretrained the AlexNet with the train data of the Photo domain, using the train split introduced in the original paper (Li et al., 2017). We pretrained the model for 60 epochs, with a learning rate of $5e-3$ using the SGD optimizer. We further used the Step learning rate scheduler with a gamma rate (i.e. the strength of the learning rate decay) of 0.5. The batch size was set as 32.

**Corrupted CIFAR-10**  For the corrupted CIFAR-10 experiment, we pretrained the WRN with the train split of CIFAR-10. The pretraining epochs was set as 200, with a learning rate of $1e-1$ using the SGD optimizer. We used the Multi-Step LR scheduler, setting the gamma rate as $2e-1$, with milestones set as $\{60, 120, 160\}$. Hence, every time the training epoch reaches the milestone, the learning rate was reduced to one-fifth of the previous rate. The batch size was set as 128.

**Digits**  Lastly, for the Digits experiment, we set the number of pretraining epochs as 100, with a learning rate of $1e-4$ using the Adam optimizer. The batch size was set as 256.

**Office-Home**  We pretrained the ResNet18 with the train split of the Real World domain. We pretrained the model for 100 epochs, with a learning rate of $1e-4$ using the SGD optimizer. We used no learning rate scheduler. The batch size was set as 64.

## B.4 HYPERPARAMETERS

In this part, we state the hyperparameters used in our experiments.

$\lambda_{\mathbf{PROF}}$  $\lambda_{\text{PROF}}$ is a balancing coefficient for $L_{\text{PROF}}$, an objective adopting the feature-decorrelation loss introduced in Zbontar et al. (2021). We tuned $\lambda_{\text{PROF}}$ using experimental results of the original paper and (Tsai et al., 2021). In the original paper, the author reported the optimal value of the balancing term as 0.005, which remains consistent under varying projection dimensions. We set this as a starting point for hyperparameter tuning. We find that if $\lambda_{\text{PROF}}$ balances the off-diagonal term (i.e. redundancy reduction term) and the diagonal term (i.e. alignment term) to a similar degree, no significant differences are observed. Furthermore, switching $\lambda_{\text{PROF}}$ to $\frac{1}{d} \approx 0.0001$ showed no significant changes to the learning process. Here, $d$ denotes the projection dimension of the distillation head $\mathcal{V}$ (distillation head output space). While we cannot guarantee an optimal value for $\lambda_{\text{PROF}}$, we set $\lambda_{\text{PROF}} = 0.005$ for our two experiments using PROF.

$\lambda_{\mathbf{MDAR}}, \lambda_{adv}$    The hyperparameters $\lambda_{\mathrm{MDAR}}$ and $\lambda_{adv}$ is used together for adversarial learning, hence we report the two together. $\lambda_{\mathrm{MDAR}}$ was set in a similar way as $\lambda_{\mathrm{PROF}}$. For our experiments, $\lambda_{adv}$ was set as 0.005. $\lambda_{adv}$ was searched under a fixed value of $\lambda_{\mathrm{MDAR}} = 0.005$. We experimented with varying values of $\lambda_{adv}$: $\{0.005, 0.05, 0.5\}$, which showed no significant difference to the training process, while 0.05 showed slightly better results in the validation set of the source domain. Hence, in our experiments, $\lambda_{adv}$ was set to 0.05. To explicate, generally, $L_{adv}$ displayed a value approximately 10 times larger than $L_{\mathrm{MDAR}}$. We believe that this behavior is correlated to 0.05 being a good value for $\lambda_{adv}$ under a fixed value of $\lambda_{\mathrm{MDAR}} = 0.005$.

All other hyperparameters (e.g., $w_{cyc}, w_{div}, w_{adv}, w_{\mathrm{PROF}}$ ) are searched with a similar method to Li et al. (2021). For all experiments, we set $w_{cyc}$ as 20.0, $w_{cyc}$ as 2.0, and $w_{adv}$ as 0.1 in Digits, and 0.02 in PACS and Corrupted CIFAR-10. Finally, $w_{\mathrm{PROF}}$ was set as 0.1. The values were tuned such that the weighted losses (i.e. $wL$) are situated in a similar range.

## C   ON DOMAIN GAPS

In previous works, there exist different mentions regarding the domain gap within the experimental datasets. We begin this section by comparing such views.

There are contradicting views on the domain gap within the PACS dataset, the authors of Wan et al. (2022) view that the domain gap is significant between the Art domain and the source domain (Photo), while relatively smaller with the Sketch and Cartoon domain. In contrast, Wang et al. (2021) viewed that the domain gap is the largest between the source and the Sketch domain, due to its vastly abstracted shapes. On the contrary, there exists a shared consensus regarding the domain gap within corrupted CIFAR-10 dataset, where researchers view that the domain gap between the source (CIFAR-10) and the target (corruption datasets) is defined by the severity level of the corruption (Li et al., 2021; Qiao et al., 2020; Wang et al., 2021; Wan et al., 2022). Concerning the Digits dataset, the authors of Qiao et al. (2020); Wang et al. (2021); Li et al. (2021) view that USPS displays the smallest domain gap with the source domain (MNIST). This is very similar to the view of Wan et al. (2022) that USPS and SYNDIGIT datasets are closer to the source, while there is a large domain gap between the MNIST-M and the source domain.

In our paper, we used a different measure to observe the domain gap between datasets: the OOD classification accuracy on unseen domains. Our view on domain discrepancy is that it can be indirectly observed through the downstream task performance. This is closely tied to realistic settings, where task performance is the leading motive behind the study of sDG. The method is simple: using a fixed model, we train the model with the train split of the source domain. Then, using the trained model, we test the classification accuracy on unseen domains. We reported the results in Section 5.2. Using the baseline OOD accuracy as a measure for domain gap matches the view of many existing works, while differences exist. For instance, USPS displays the highest OOD accuracy, matching the view of previous works that USPS shows the smallest discrepancy with the source (Qiao et al., 2020; Wang et al., 2021; Li et al., 2021; Wan et al., 2022). In PACS, the Sketch domain displays the lowest baseline OOD accuracy, which is in line with the view of some previous works (Wang et al., 2021), while different from the view of Wan et al. (2022).

## D   ON ORACLES

In this section, we discuss the implementation of the oracle using pretrained models. Using pretrained models for OOD generalization is not an entirely novel idea (Li et al., 2023; Cha et al., 2022), but first for the task of sDG.

We selected the pretrained RegNetY-16GF as an oracle for PACS. In Cha et al. (2022), a pretrained RegNetY-16GF model displayed high MI with the true oracle, a model that is trained on all source and target domains). The authors reported that the true oracle displayed an average validation accuracy of $98.4\%$ on all PACS domains.

Similar to this, our implementation of the oracle with a pretrained RegNetY-16GF finetuned on the source domain (i.e. Photo in PACS, MNIST in Digits, Real World in Office-Home) displayed high validation accuracies across all target domains. To be specific, in PACS, the finetuned RegNetY-16GF marked $75.16\%$, $75.30\%$, $69.00\%$ on Art, Cartoon, Sketch, and an average validation accuracy

of 73.15. While the average accuracy is lower than the true oracle in Cha et al. (2022), this is an expected behavior as our oracle used only the Photo domain, while the true oracle in (Cha et al., 2022) utilized all four domains of PACS.

However, we empirically confirm that the RegNetY-16GF is not universally available for use as the oracle. For instance, using the RegNetY-16GF to implement the oracle for the Corrupted CIFAR-10 experiment was not satisfactory. When finetuned with the source domain (i.e. CIFAR-10), RegNetY-16GF marked low validation accuracy in the target domain with an average of $60.65\%$. This is similar for the implementation with ResNet50, which marked an average accuracy of $61.25\%$ on the target domains, performing worse than the task model. We believe that this difference is derived from the difference between the two datasets. For instance, PACS is a collection of images without any distortion, while the Corrupted CIFAR-10 is a dataset generated by vastly distorting CIFAR-10. As the RegNetY-16GF is not specifically trained to withstand distortions, its performance decrease in Corrupted CIFAR-10 is understandable. Similarly, the RegNetY-16GF does not fit well with the Digits benchmark due to the large gap between the pretrained dataset of the RegNetY-16GF and the Digit classification datasets.

This issue can be explained with the work of Wolpert & Macready (1997), where the authors demonstrate that there exists a trade-off between a model's performance on a certain task and the performance on all remaining tasks. We believe this to be a crucial limitation of our method, and aspire to investigate further.

