# OpenReview forum: "Learning to ignore: Single Source Domain Generalization via Oracle Regularization"
_ICLR.cc/2024/Conference — Submitted to ICLR 2024_

### Official Review · Reviewer_N8Mf · 2023-10-22

**Soundness:** 2 fair
**Presentation:** 2 fair
**Contribution:** 2 fair
**Rating:** 3
**Confidence:** 5

**Summary:**

This paper presents a method called Progressive mutual information Regularization for Online distillation of Frozen oracles (PROF) for addressing single-source Domain Generalization (sDG) under large style domain shifts. A popular approach is to learn robustness via the alignment of augmented samples. However, prior works frequently overlooked what is learned from such alignment. To address this, the authors introduce PROF, a novel regularization method that leverages pretrained models (oracles) to guide the learning process via a feature-level regularization of mutual information. The authors also present an alignment objective, Multi-Domain Alignment with Redundancy Reduction (MDAR), that serves as a strong sDG baseline. The paper is focusing on style domain shifts (but not others).

**Strengths:**

1. The paper tackles an essential research problem of single-source Domain Generalization for style domain shift, a topic still awaiting further exploration.

2. The overall organization of the paper is relatively clear. However, the method description section is not well-organized (see weaknesses).

**Weaknesses:**

1. Organization and Clarity in Section 4.

The method description in Section 4 is not reader-friendly. Generally, when a final loss function consists of multiple loss terms, it's more intuitive to first introduce each term separately, explaining their purpose and how they interact, before presenting them as a sum. However, in this section, the authors introduce the summation of loss items before elaborating on each term, which may confuse readers. In particular, The term L_{cls} in equation (2) is not introduced anywhere either in words or in maths before equation (2), and only clarified in Section 4.3. This organization makes understanding the method unnecessarily difficult and time-consuming, even for readers familiar with single-source DG methods like me. Additionally, the symbol "D" is used to represent different concepts in Sections 3 and 4.

2. Incomplete Reference to Related Works and Baselines.

 There are numerous regularization-based strategies for DG in the literature that haven't been addressed in this paper. For example, Probability matching is employed in ReLIC [1] and AugMix [2]; logit matching is used in CoRE [3]; and feature matching is deployed in MatchDG [4]. However, these works are neither discussed nor evaluated in the paper.

[1] Jovana Mitrovic, Brian McWilliams, Jacob C Walker, Lars Holger Buesing, and Charles Blundell. Representation learning via invariant causal mechanisms. In ICLR, 2021.

[2] Dan Hendrycks, Norman Mu, Ekin D. Cubuk, Barret Zoph, Justin Gilmer, and Balaji Lakshminarayanan. AugMix: A simple data processing method to improve robustness and uncertainty. Proceedings of the International Conference on Learning Representations (ICLR), 2020.

[3] Christina Heinze-Deml and Nicolai Meinshausen. Conditional variance penalties and domain shift robustness. Machine Learning, 110(2):303–348, 2021.

[4] Divyat Mahajan, Shruti Tople, and Amit Sharma. Domain generalization using causal matching. In International Conference on Machine Learning, pp. 7313–7324. PMLR, 2021.

3. Lack of Comprehensive Ablation Study.

The loss function of the task model, as presented in equation (2), comprises several loss terms, each containing multiple sub-components. Without a thorough ablation study, it's challenging to understand the contribution of each sub-component towards the overall DG performance or to verify that the combination proposed by the authors is indeed the most effective.

Specific questions that arise from this include, but are not limited to:

- What would the results be if the last two terms in equation (2) were removed, i.e., simply using the examples generated by the generator alongside the original examples in ERM training?
- How does the oracle model alone perform on the DG task? Given that the oracle model is trained on a much larger dataset, it may already see images in varying styles. Is the DG performance improvement in PROF mainly due to domain leakage from the oracle model?
- How crucial are D and P, and the Barlow Twins in equations (3) and (5)? What would the results look like if simple feature matching/logit matching/probability matching were used instead?
- What would be the impact of using a different number of augmented views in equation (5)?
- Could the authors provide more rationale behind the current generator's design? Is there a specific reason it must be designed this way? How would the results be affected if a different generator was used?

These questions highlight the need for a more detailed ablation study to fully understand and validate the proposed method. Without such an analysis, it's reasonable to question the necessity of each component and specific design.

4. The performance of the proposed method in the dataset with style shift in the more real-world scenario, such as the FMoW-WILDS and Camelyon-WILDS (https://wilds.stanford.edu/) is unknown.

In summary, the unclear method description, the absence of several related works and baselines, and the lack of a comprehensive ablation study led me to lean toward rejecting this paper. However, I am open to further discussion and may reconsider my decision based on the authors' responses.

**Questions:**

See weaknesses above.

**Details Of Ethics Concerns:**

N/A.

---

> ### Author Response · Authors · 2023-11-20
> **Response to Reviewer N8Mf**
>
> We appreciate the reviewer for spending your valuable time reviewing our work. We thank the reviewer for commenting that the organization of our paper was clear. However, we feel that there is room for improvement. Below, we carefully address your concerns regarding our work.
>
> ***
> # Organization and Clarity in Section 4.
>
> - **We provide a revised version that reflects the reviewer’s concern.** Thank you for providing meaningful suggestions on the writing. We will make sure your suggestion regarding the definition of L_cls is well addressed in advance. Regarding the symbol D, we believed that the domain D (Section 3), and the distillation head D (Section 4) are considerably close, as distillation head D functions as a way to mitigate the issue derived by the effect of the domain variable.
> ***
> - **We have changed notations, references, and figures.** Nonetheless, we reflected the reviewer’s concern and changed the term of the distillation head from D to V throughout the paper. Please refer to Figure 2, and notations. We also added a proper reference to L_cls for the readers. *The changed lines are marked in blue*.
>
> ***
> # Incomplete Reference to Related Works and Baselines.
>
> - **We have carefully selected preliminary works that best fit with our research.** Thank you for providing us with related papers that leverage the concept of invariance, or use causal models. However, as mentioned in the introduction, the single-domain setting of the sDG task makes conventional approaches in DG simply unavailable due to the unobservability of the domain D.
> ***
> -  **Furthermore, we believe that not all papers are necessarily related to our case.** For instance, Mitrovic et al. (2021) is a paper that belongs to the field of self-supervised learning. The paper uses a causal graph, yet does not provide identifiability results available for use in our case. Mahajan et al. (2021) is a paper on DG, yet not on sDG. Hence, the approach of Mahajan et al. (2021) does not fit our setting.  Regarding the other papers, we are grateful for the mention and are happy to add them to our related works in Section 2, *marked as blue*.
>
> ***
> [1] Khosla et al. Supervised Contrastive Learning, 2021.
> \
> [2] Park et al., Relational Knowledge Distillation, 2019.

---

> ### Author Response · Authors · 2023-11-20
> **Response to Reviewer N8Mf (continued)**
>
> Below, we continue our response
> ***
> # Ablation Study
>
> > "What would the results be if the last two terms in equation (2) were removed, i.e., simply using the examples generated by the generator alongside the original examples in ERM training?"
> - **Alignment is helpful in increasing both ID and OOD performance. We include a new experiment result to show this.** Using simply the cross-entropy loss, the OOD performance drops. In Digits, the average score is 68.10. This is significantly lower than both PROF(81.82) and our baseline MDAR(78.80) The effect of using the augment & alignment method was studied earlier in the sDG literature and the field of self-supervised learning(e.g., Supervised Contrastive Learning [1]). Empirically, we add an experimental result on the effect of alignment.
> - Without Alignment, the OOD generalization score is: mnist_m: 73.06/ usps: 81.56/ svhn: 56.47/ syndigit: 62.35
>
> ***
> > "How does the oracle model alone perform on the DG task? ... Is the DG performance improvement in PROF mainly due to domain leakage from the oracle model?"
> - **Information on the performance of oracles is included in Appendix D. On Oracles.** We utilized pretrained models which are expected to exist in many real-world cases. The oracle we used (i.e., RegNet-Y-16GF) is pretrained on the Instagram dataset.
> - **We believe that domain leakage has not taken place.** If there were a domain leakage, the OOD results should be much higher than what we have presented. This can be inferred from the result in Table 3, where PROF outperformed conventional methods by a large gap.
>
> ***
> > "How crucial are D and P, and the Barlow Twins in equations (3) and (5)? What would the results look like if simple feature matching/logit matching/probability matching were used instead?"
> - **The role of projection heads is a long, studied topic in the field of self-supervised learning.** To us, the role of the projection head was significant as the cost of directly aligning the feature outputs of H_o and H (two vectors of length 3024) is immensively expensive. We have also tested the method of relational knowledge distillation [2], but could not produce similar results as PROF. We have not used logit matching methods, as we did not aim to use the classification layer, as it is fitted to the original task.
> - **Barlow Twins is very helpful in converging the alignment loss**, mainly because it works well on small batch sizes, hence fit for our cases such as PACS.
>
> ***
> > "What would be the impact of using a different number of augmented views in equation (5)?"
> - **Raising the number of augmented views in the alignment optimization affects the generalizability of the task model.** With its optimum being 3~4. This was originally studied in the field of self-supervised learning, where increasing the number of augmented views significantly helps performance ( Please refer to Section 11. of Khosla et al. (2021) [1]), However, raising this number increases the cost, with very marginal gains in OOD generalizability after a certain point.
>
> ***
> > "Could the authors provide more rationale behind the current generator's design? Is there a specific reason it must be designed this way? How would the results be affected if a different generator was used?"
> - **The generator design follows the work of Li et al. (2021).** As mentioned in the paper, the generator design follows the work of Li et al. (2021). The authors presented a progressive domain expansion method via augmentation. This method fits our case as we aim to mitigate issues derived by repetitive augmentation.
>
> ***
> [1] Khosla et al. Supervised Contrastive Learning, 2021.
> \
> [2] Park et al., Relational Knowledge Distillation, 2019.

---

> ### Comment · Reviewer_N8Mf · 2023-11-23
>
> Thank you for your response. However, upon careful examination of your response, I find that my initial major concerns remain:
> - The quality of writing in Section 4.1 still requires substantial improvement.
> - In accordance with the concerns voiced by Reviewer TGW3: The necessity for each component in Figure 2 is not sufficiently justified.
>
> Therefore, I will keep my score.

---

### Official Review · Reviewer_6mfm · 2023-10-31

**Soundness:** 2 fair
**Presentation:** 2 fair
**Contribution:** 2 fair
**Rating:** 3
**Confidence:** 3

**Summary:**

The paper critiques augmentation-based Single-source Domain Generalization (sDG) methods, emphasizing the distinction between domain and augmentation invariance. To tackle this, this work leverages pre-trained models for mutual information regularization.

**Strengths:**

- The paper takes a deep dive into augmentation-based sDG methods and highlights existing challenges. Analyzing the data-generating process and the distinction between domain invariance and augmentation invariance provides insights.
-  The introduction of the PROF method for feature-level regularization of mutual information leads to improvements over several benchmark datasets.

**Weaknesses:**

- The use of pre-trained models for "oracle" regularization in the domain generalization context is not entirely novel, as evidenced by prior works such as [1]. From a technical standpoint, the advancements proposed here appear to be incremental.

- While the critique of sDG methods and the subsequent motivation for the proposed method is commendable, mere verbal analysis falls short of expectations. There appears to be a void in terms of rigorous theoretical groundwork. Specifically, the paper lacks a deep theoretical analysis, such as the provision of a generalization bound, that sheds light on the functioning and merits of PROF.


[1] Cha, J., Lee, K., Park, S., & Chun, S. (2022, October). Domain generalization by mutual-information regularization with pre-trained models. In European Conference on Computer Vision (pp. 440-457). Cham: Springer Nature Switzerland.

**Questions:**

See weakness.

---

> ### Author Response · Authors · 2023-11-20
> **Response to Reviewer 6mfm**
>
> Thank you for your time reviewing our work. We appreciate you for pointing out that we provided a "distinction between domain invariance and augmentation invariance", which is a phenomenon rarely noticed in the field of sDG in general. Below, we have addressed your concerns regarding our work. Thank you.
>
> ***
> # Weaknesses
>
> > "From a technical standpoint, the advancements proposed here appear to be incremental (e.g., MIRO)."
> - **MIRO and PROF originate from a differing motive.** We acknowledge that the idea of oracle regularization was introduced earlier in the DG literature (while not on the sDG literature). This is why we referenced the paper in Section 2. However, the motivation of the two papers strongly differ. MIRO is designed under the consideration that learning domain-invariance from multiple domains is uncertain. On the other hand, the motive of PROF derives from a causal notion that augmentation cannot assure robustness under large domain shifts.
>
> ***
> > "Specifically, the paper lacks a deep theoretical analysis, such as the provision of a generalization bound, that sheds light on the functioning and merits of PROF."
> - **We provide an analysis from the causal perspective, using the insight of a previous work from a different domain.** The focus of our work lies in approaching the generalization problem from a causal perspective, focusing on the effect of data augmentation. Our work adopts the identification process of the causal model from the work of Kugelgen et al. (2021) [1] but provides a new perspective on the causal model with the inclusion of the domain variable D. This is elaborated in Section 3. A causal interpretation of data augmentation.
>
> ***
> [1] Kugelgen et al., Self-supervised learning with data augmentations provably
> isolates content from style, 2021.

---

> > ### Comment · Reviewer_6mfm · 2023-11-23
> > **Thanks**
> >
> > Thanks for your efforts in addressing my concerns. I decided to keep my original ratings as my major concern about the technical contribution of the paper still remains.

---

### Official Review · Reviewer_DsCE · 2023-11-01

**Soundness:** 2 fair
**Presentation:** 2 fair
**Contribution:** 2 fair
**Rating:** 5
**Confidence:** 3

**Summary:**

In this paper, the authors suggest that conventional augmentation-based single source domain generalization methods cannot effectively handle large domain shift scenarios. They study this phenomenon from the point of view of S-C disentanglement and propose a novel augmentation-based sDG baseline, MDAR. Besides, they introduce an oracle regularizer, PROF, which utilizes an oracle model and knowledge distillation method to solve this problem. Experiments on several DG benchmarks verified the severe OOD performance fluctuations.

**Strengths:**

- The paper is well-organized.

- The related work and the background knowledge are comprehensive and overall complete.

- Discussions and questions about the quality of data augmentation and the effects of alignment are insightful.

**Weaknesses:**

- There are some presentations that could be more precise. Please refer to the Questions.

- The novelty of this article is limited. The proposed network components and corresponding losses are a little redundant. Lack of discussion of the necessity for each component.

- The experimental results, compared with some SOTA performances, are not very convincing.

**Questions:**

- Presentations:

    - In the abstract, it is mentioned “analyzing the data generating process“. But, I did not find any explicit explanations in the main paper.

    - Is there any formal explanation or computational and mathematical definition of mid-train fluctuations?

    - For “Learning to ignore” under section 3, there are some incomplete phrases. Besides, a clearer explanation of why a less severe mid-train fluctuation means a better capture of domain invariant features should be added.

- Experiments:

    - The experimental results and the analysis mention that the proposed PROF regularizer only performs well under some large domain shift scenarios. Will this limit the application and the contribution of the proposed method?

    - What is the difference or the benefit of the proposed baseline MDAR, compared with other augment and align methods?

    - As the PROF aims to guide the task model to learn more domain-invariant knowledge from the oracle model, I think a study of different oracle models’ influence is necessary.

    - While measuring the mid-train fluctuations, are the results an average of multiple experiments or a single experiment? Is it statistically significant?

---

> ### Author Response · Authors · 2023-11-20
> **Response to Reviewer DsCE**
>
> Thank you for your time and for reviewing our work. We especially thank you for noting that we gave an insightful discussion on “data augmentation" and the "effect of alignment". While data augmentation is commonly considered a promising method for sDG, a deep analysis of its underlying mechanism has been absent. Revealing the overlooked uncertainty of data augmentation was one of our core research objectives. Down under, we have addressed your concerns regarding our work.
>
> ***
> # Presentations
> > "In the abstract, it is mentioned “analyzing the data generating process“. But, I did not find any explicit explanations in the main paper."
> - Regarding your note on the absence of explicit analysis on “the data generating process”, we ask you to refer to Section 3. “A causal interpretation of data augmentation” on page 3.
>
> ***
> > "Any formal explanation or computational and mathematical definition of mid-train fluctuations?"
> - **The notion of mid-train fluctuation is a relatively new topic.** The phenomenon is also mentioned in Arpit et al., (2022) [1], but not in the field of single source domain generalization. While a mathematical definition of mid-train fluctuation is itself a new research topic, we can easily observe its presence. For instance, the OOD performance varies greatly during the training of a model.
>
> ***
> > "clearer explanation of why a less severe mid-train fluctuation means a better capture of domain invariant features"
> - **Mid-train fluctuation is an issue that should be mitigated.** Thank your interest on the relation between mid-train fluctuation and domain-invariance. The phenomenon of mid-train fluctuation is a common phenomenon observed in both DG and sDG papers. In DG, Arpit et al., (2022) [1] viewed this fluctuation as a sign of difficulty in model selection. We view mid-training fluctuation as a serious issue since it manifests that the simulated domains do not properly reflect unseen domains and, further, it harms the credibility of learned models due to uncertainty in their real-world performance
> - Our hypothesis is that the fluctuation derives from the failure in approximating domain-invariance via augmentation-invariance. Under this assumption, we design PROF, a regularization method to ensure the learning of domain-invariant representations from a domain-agnostic generalized model.
>
> ***
> # Experiments
>
> > "The experimental results and the analysis mention that the proposed PROF regularizer only performs well under some large domain shift scenarios. Will this limit the application and the contribution of the proposed method?"
> - **PROF can be used for both large/small distribution shifts, but intended for large shifts.** We believe that for small domain shifts (e.g., image corruption), data augmentation is sufficient. However, this approach would not work between large domain shifts. Intuitively, it is not easy to imagine augmenting a photo of a dog with a cartoon image of a dog. While recent progress in generative models may help this issue, such methods are costly, and hence not readily available for use.
>
> ***
> > "What is the difference or the benefit of the proposed baseline MDAR, compared with other augment and align methods?"
> - **MDAR is a strong baseline. We use it for fair comparison against PROF.** We design MDAR as a fair baseline against PROF. MDAR works well on small batch sizes as it uses a non-contrastive alignment loss [2]. Hence compared to conventional methods, MDAR converges better than conventional alignment objectives (e.g., Supervised Contrastive Loss).
>
> ***
> > "I think a study of different oracle models’ influence is necessary."
> - **We have investigated different oracles.** We showed that for general cases,  a pretrained model trained on photo images (e.g., RegNet-Y-16GF) is effective for oracle regularization. Naturally, such a model may not work for digit classification tasks. The extensive analysis of oracles is reported in Appendix D. *On Oracles*.
>
> ***
> > "While measuring the mid-train fluctuations, are the results an average of multiple experiments or a single experiment? Is it statistically significant?"
> - **Mid-train fluctuation is prevalent, and PROF repetitively stabilizes it.** The mid-train fluctuations are the results of individual experiments, but we observe that OOD fluctuation exists on every run. This matches the fluctuations reported in Arpit et al., (2022) [1]. We view this issue as a significant threat to the credibility of the model.
>
> ***
> [1]  Arpit et al., Ensemble of averages: Improving model selection and boosting performance in domain generalization, 2022.
> \
> [2] Tsai et al., A Note on Connecting Barlow Twins with Negative-Sample-Free Contrastive Learning, 2021.

---

### Official Review · Reviewer_TGW3 · 2023-11-03

**Soundness:** 2 fair
**Presentation:** 1 poor
**Contribution:** 1 poor
**Rating:** 3
**Confidence:** 3

**Summary:**

The paper studies Single Source Domain Generalization where the models are only allowed to train on images from a single source and should generalize to other domains. The authors highlight some issues and failure modes with prior works based on augment-and-align, and introduce a feature-level regularization method via distillation to improve outcomes.

**Strengths:**

The paper attempts to improve performance on a relevant problem – single-domain generalization. Presumably, methods that work well here will give deeper insight into feature learning & generalization.

**Weaknesses:**

The combination of a) a complex, not clearly motivated solution, b) lack of clarity on key contributions and empirical evidence to back this up (lesion analysis), c) overall weak results make this paper fall a bit short of the mark.

* Clarity: Although the authors seems to have put a lot of effort in explaining different components being used in the method, I believe there is a lot of scope of improvement.
    * Currently, it’s very difficult to follow and understand what, precisely, is the problem being solved, why we should care, and what is the exact novel idea being contributed. Sections 2 & 3 introduce a number of concepts (causal graphs, augmentation, training dynamics, distillation, etc., all from previous work, but do not really help clarify e.g., why these observations should form the logical starting point of a system for single source domain generalization.
    * Figure 2 illustrates the system which has a surprisingly large number of components – ensembling of generators, augmentation schemes, distillation from an oracle, an oracle regularizer, again many being adapted from previous works. There’s also the use of a Multi-domain alignment loss. It’s not clear why this specific combination of components is useful, which ones are new, and how/why these pieces work together.

* Results:

    * The results are mixed at best, with e.g., table 2 “Ours” not being any better than metaCNN on any column (but bolded in multiple places incorrectly), table 3 being quite variable too, and table4 having no baseline comparisons.
    * Lacking explanations: Although authors motivate the paper by claiming that PROF is better than augment-and-align as it reduces mid-training OOD fluctuations, they fail to explain the failure modes (eg training domain: Cartoon, Target Domain Sketch: 0.70 -> 3.91). More study is required to properly understand when the proposed method becomes unstable.
    * As mentioned in the paper, the proposal of using oracle model to minimize mutual information with it is not novel and is already proposed in MIRO[1], so it would make sense to directly compare against it.

[1] Domain Generalization by Mutual-Information Regularization with Pre-trained Models

**Questions:**

please see above.

---

> ### Author Response · Authors · 2023-11-20
> **Response to Reviewer TGW3**
>
> # Response to Reviewer TGW3
>
> Thank you for the thoughtful and encouraging feedback. Thank you for noting that the paper provides "deeper insight into feature learning & generalization". We believe that a causal perspective opens new room for discussion in the field of generalization. We have addressed your concerns below.
>
> ***
>
> # Response on Weakness(Clarity)
>
> > "it’s very difficult to follow and understand what, precisely, is the problem being solved, why we should care, and what is the exact novel idea being contributed"
>
> - **We answer the question: Is augmentation reliable for generalization?** To our understanding, the focus of the sDG task lies in designing effective data augmentation techniques to improve model robustness. However, existing works on sDG often overlook the discussion on the safety of using augmentation for robustness. Our work questions this very point, “Is augmentation reliable for generalization?”
> ***
> > "why these observations should form the logical starting point of a system for single source domain generalization"
>
> - **We illuminate an overlooked uncertainty of data augmentation using causality.** To answer this question, we start from a causal perspective. The causal examination aims to understand the underlying mechanism of data augmentation. Using a causal model, we reveal that data augmentation in the sDG setting does not promise domain-invariance. While we adopt the results of Kugelgen et al. (2021) [1], we are different from their work in the sense that we add an additional variable into the model, the domain variable D. To the best of our knowledge, this is the first work to utilize causal models to reveal the uncertainty of augmentation-based sDG.
> ***
> > "Figure 2 illustrates the system which has a surprisingly large number of components"
>
> - **Our data augmentation module is intended to observe the long-term effect of data augmentation on generalizability.** As mentioned in the second line of Section 4.3., the aim of the complex data augmentation module is to allow repetitive augmentation. Unlike random data augmentation, our adversarial augmentation creates augmented views that are supposedly helpful for OOD robustness. Nonetheless, even in this case, the phenomenon of OOD fluctuation is present. As for now, PROF is the first method to significantly mitigate such issues. In a similar work of Arpit et al., (2022) [2], the authors used an ensemble of models to mitigate the fluctuation. Ours produces the stabilization effect with a single model.
>
> ***
> # Response on Weakness (Results)
>
> > "The results are mixed at best"
> - **Generally, PROF stabilizes the phenomenon of OOD fluctuation, fulfilling its main objective.** We hope to underline that the focus of our method lies in mitigating the uncertainty of augmentation, revealed as the OOD fluctuation phenomenon. To be best of our knowledge, PROF is the only method that directly targets this problem. In this sense, PROF fulfills its objective in the stabilization of OOD performance.
> ***
> > "table 4 having no baseline comparisons"
> - **The office-home dataset is not a conventional dataset for sDG. But Why?** The Office-Home dataset(Table 4) is not a conventional dataset in the field of sDG. As there are no reported baselines, we used our baseline (MDAR) for comparison. In terms of both OOD fluctuation and generalizability, PROF outperforms the baseline. We hence derive a question: If augmentation can mitigate the problem of distribution shifts, why is it not effective under large distribution shifts?
> ***
> > "they fail to explain the failure modes"
> - **The failing cases do not weaken our logic.** As mentioned in Appendix D, we recognize that there are limitations in our method PROF. PROF relies on the idea of a generalized oracle model. From this perspective, our hypothesis on the failing cases is that the pretrained models that we approximate as oracles (e.g., RegNet-Y-16GF) are originally trained on real-photo images. We expect that if the oracle is also trained with cartoon or sketch images, it would show no defects in the reported failing cases.
> ***
> >"the proposal of using oracle model to minimize mutual information with it is not novel and is already proposed in MIRO"
> - **MIRO and PROF each originate from a differing motive.** We acknowledge that the oracle regularization was introduced earlier in the DG literature (as mentioned in Section 2. Preliminaries).  However, the motivation of the two papers strongly differ. MIRO is designed under the consideration that learning domain-invariance from multiple domains is uncertain. On the other hand, the motive of PROF is rooted in the notion that augmentation cannot assure robustness under large domain shifts, supported by a causal analysis.
>
> ***
> [1] Kugelgen et al., Self-supervised learning with data augmentations provably
> isolates content from style, 2021.
> \
> [2]  Arpit et al., Ensemble of averages: Improving model selection and boosting performance in domain generalization, 2022.

---

> > ### Comment · Reviewer_TGW3 · 2023-11-23
> > **Response to author rebuttal**
> >
> > Dear authors,
> >
> > Thank you for the time and effort invested in sharing a detailed response.  Having read carefully through your response, I am unfortunately not moved to change my assessment significantly. The primary concerns remain unchanged in large part. 1) unexplained complexity: why do we *need* all the bits in figure 2, which are new, how much do they contribute; 2) weak results: the shared results do not show significant, consistent improvement on substantial datasets against relevant baselines, 3) clear, writing that convincingly articulates (1,2) to readers.
> >
> > (As an aside, I continue to find the bolding in the tables perplexing -- in many columns the not-largest number is bolded, leading to confusion. The authors should fix this in future submissions).

---

### Meta-Review · Area_Chair_bzeb · 2023-12-14

**Metareview:**

The paper studies an overall interesting problem. However, the paper in its current form has several key areas of improvement as detailed in all the reviews. The reviews should serve as a useful starting point in revising and improving this paper for the next round. The paper seems to lack clarity (lot of moving parts, poorly substantiated verbal claims), complex approach (unclear which components are actually necessary, and more complex systems are less likely to reliably work in general), lack of significant empirical gains, and novelty doesn't compensate for these weaknesses.

**Justification For Why Not Higher Score:**

The paper neither has clear empirical gains, novelty or theoretical insights. All reviewers recommended rating 5 or less

**Justification For Why Not Lower Score:**

NA

---

### Decision · Program_Chairs · 2024-01-16

Reject